# Towards a Transdisciplinary Theoretical Framework of Citizen Science: Results from a Meta-Review Analysis

Andrea Spasiano [1,2,*], Salvatore Grimaldi [3], Alessio Maria Braccini [2] and Fernando Nardi [1,4]

1. Water Resources Research and Documentation Center, University for Foreigners of Perugia, Piazza Fortebraccio, 4, 06123 Perugia, Italy; fernando.nardi@unistrapg.it
2. Department of Economics, Engineering, Society and Business Organization, University of Tuscia, Via del Paradiso, 47, 01100 Viterbo, Italy; abraccini@unitus.it
3. Department for Innovation in Biological, Agro-Food and Forest Systems, University of Tuscia, Via San Camillo de Lellis, snc, 01100 Viterbo, Italy; salvatore.grimaldi@unitus.it
4. Institute of Environment and College of Arts, Science and Education (CASE), Florida International University (FIU), 11200 SW 8th Street, CASE 450, Miami, FL 33199, USA
* Correspondence: aspasiano@unitus.it

**Abstract:** This work intends to lay the foundations for a theoretical framework of citizen science combining social and organizational implications with the support of information technologies. The proposed theoretical framework moves towards a shared and common research process between experts and citizens to deal with environmental and social challenges. The role and capacity of online communities is explored and their engagement capacity by means of web-based digital platforms supporting crowdsourcing activities. In this contribution, authors highlight the most common practices, methods and issues of citizen science approaches adopted from multidisciplinary application fields to obtain insights for designing a new participative approach for organizational studies. To reach this goal, authors illustrate the results of a systematic meta-review analysis, consisting of an accurate selection and revision of journal review articles in order to highlight concepts, methods, research design approaches and tools adopted in citizen science approaches.

**Keywords:** citizen science; participatory approaches; online communities; public engagement; organizational model



## 1. Introduction

Citizen science is a set of science-driven methodological approaches that engage citizens and the general public in scientific research activities [1–5]. Citizen science programs are generally transdisciplinary, allowing multiple disciplines to join efforts with citizens and stakeholders to solve societal challenges cross-fertilizing various areas of scientific research [6,7].

Advances in digital, web-based and mobile technologies recently brought new stimuli and opportunities linked to data collection and analysis that lie on volunteers' participation [2,8–11]. It is well recognized, the scaling potential of digital and web-based technologies to enlarge the communities' and volunteers' participation, making easier the collection and analysis of data from citizens' contribution, by reducing cost of data collection [12] and guarantying large spatio-temporal coverage [2]. In this perspective, online interactions through social media platforms can enhance public participation within the theoretical framework of online communities as new potential actors of citizen science approaches. Volunteered users' interactions may contribute to create large database, address the lack of information and support large-scale surveys and interviews [13,14].

Citizen science is a research approach derived from ecological, environmental and earth observation sciences [15]. Hydrology, water resources management and land use/urban planning represents major fields of application of citizen science [2,16] to address

data scarcity and build innovative analytical models. The integration of computer science—which investigates into technical issues—and social science [13,17]—is going to broaden citizen science investigation on motivational aspects, human behavior and socio-economic implications [16]. In this way, citizen science is not just an auxiliary methodology, but is going to become a cross-cutting discipline [2] with characteristics of a distinct field of inquiry [3].

From an experimental approach to sectoral scientific research, citizen science is gradually evolving towards large-scale application models, aimed at integrating methods and tools from different disciplines. This evolution involves not only conceptual and epistemological changes, but also the organizational level. The latter concern aspects such as the engagement of volunteers and the stakeholder within organized communities to support scientific activities, the assignment of roles and tasks within voluntary organizations, and the extension of research goals towards cognitive, behavioral and organizational issues.

Despite the recent and rapidly increasing heterogeneous and vast set of citizen science applications, a theoretical framework is still missing. Application of citizen science in the transdisciplinary field seems to reflect empirical or experimental investigation approaches, limited to specific research areas, rather than being based on established theoretical models. Scientific efforts oriented towards the definition of the theoretical framework of citizen science are found in addressing hydrological challenges [16]. The Citizens AND HYdrology (CANDHY) Working Group—established by the International Association of Hydrological Sciences (IAHS)—is engaged in conceptualizing a transdisciplinary framework for valuing citizen science in hydrological science by promoting joint efforts with computer and social sciences. CANDHY's joint efforts aim at integrating the human and behavioral mechanism into its framework [16].

The need to combine the social and technological into citizen science approaches offers new insights for modelling processes in scientific research and for investigating organizational changes in participatory approaches. However, these aspects are not systematized within a replicable and universally applicable theoretical framework.

The lack of a transdisciplinary theoretical framework constitutes a research gap in the topic of citizen science. In order to reduce this gap, this work aims to review concepts, tools, organizational and engagement issues, limits and opportunities deriving from multidisciplinary application of citizen science. This contribution takes origin, therefore, from the need to provide theoretical insights for building and managing citizen science projects, that can be applied for designing a new and replicable participative approach. Research questions, at the base of this work, are: What essential characteristics distinguish citizen science? What insights can be grasped to define a theoretical framework applicable at the transdisciplinary level?

To reach this goal, this paper illustrates the findings of a systematic meta-review consisting of an accurate selection and revision of journal review articles in order to summarize the most recurrent concepts, methods, research design approaches and tools related to citizen science.

This contribution is structured as follows: Section 2 introduces the adopted research design in the literature meta-review analysis, by presenting work methodologies and its results. Section 3 illustrates findings of meta-review analysis divided in general definitions and common issues on citizen science practices; volunteers' recruitment and engagement; tools and data; outcomes in terms of socio-economic benefits; limits and challenges. Section 4 is focused on discussion in which authors describe the significance of findings. Section 5 is concentrated on conclusions in which authors give answers to research questions.

## 2. Research Design

The research design is based on a reasoned selection of journal review articles on citizen science topics aimed to extract concepts, definitions, methods and tools from different disciplines, such as environmental studies, water resources management and

land use and resources management. In particular, journal review article selection aims to investigate aspects related to human behavior, volunteers' recruitment and engagement by the use of information systems and digital technologies in order to enhance the role of online communities into citizen science approaches.

Research questions, at the base of this work, pose reflections on specific concerns such as: (1) level of engagement from data collection activities to research design and final outcomes dissemination; (2) impacts on organizational and management issues with the support of digital and web-based technology—such as, personal (geolocated) devices, social media, apps; (3) socio-economic benefits based on people's awareness and knowledge around common concerns.

Focusing attention to these three specific concerns, authors firstly defined a research query into the database index, following a Title–Abstract–Keywords selection. Two indexing databases were selected for this purpose: SCOPUS and Web of Science. In the latter case, the query was carried on through Topic selection, the equivalent of the Title–Abstract–Keywords selection on SCOPUS. Considering the vast number of scientific publications on the topic, authors set work as a meta-review analysis, by selecting and analyzing only review articles in order to point out common definitions, concepts, methods and technologies that characterize citizen science approaches over the disciplinary boundaries.

As illustrated in Table 1, this query considered "citizen science" as the pivotal keywords, related to a selected group of keywords concerning participatory approaches, information systems, organization studies and computer science through the Boolean operator AND. Then, authors applied a filter, including only review journal articles.

**Table 1.** Selection steps of review journal articles.

| Item | Description |
| --- | --- |
| Source | SCOPUS database—Web of Science |
| Query | TOPIC ("citizen science" AND ("participatory approaches" OR "participative approaches" OR "crowdsourcing" OR "public engagement" OR "public participation" OR "community-based monitoring" OR "participatory monitoring" OR "peer production" OR "online communities" OR "digital technologies" OR "social media" OR "human behavior" OR "motivation" OR "recruitment" OR "volunteers" OR "decision-making" OR "citizen surveillance" OR "information systems")) Refined by LANGUAGE: English and limited to DOCUMENT TYPE: Review journal paper |
| Results | 233 documents results |
| Papers retained after: <br> - No abstracts records elimination <br> Repetitive results | 164 documents results |
| Paper retained after: <br> Title and abstract selection | 68 retained documents <br> 42 uncertain results <br> 54 excluded |

Query, firstly, provided 233 results by adding results from both databases. Then, authors proceeded with a data extraction method to organize results into an Excel database. Here, authors firstly eliminated results with no authors and/or abstract. Then, authors proceeded to eliminate repetitive and redundant records, because of their presence in both database sources. Finally, a protocol of criteria was adopted for the selection of review articles. Criteria selection was structured in: (1) title and abstract selection and (2) full-text selection. In the first point, authors only selected results with the term "citizen science" among title, keywords and abstract. In fact, some results refer only to participatory approaches and, in any case, are limited to crowdsourcing or crowdfunding or aimed to support no scientific purposes. In the second point, authors gave priority to articles that investigate on recruitment and engagement issue, such as motivation, human behavior, data collection and accuracy validation protocols, tools and technologies used for citizen

science purposes. In this step, biases such as there being no review article or book chapter were excluded. In any case, some results were classified as review although they present specific cases study. Finally, authors excluded articles in which citizen science is not the paper's topic, but just mentioned as one of innovative scientific approaches, among others.

Results obtained confirm the multidisciplinary and transdisciplinary characteristics of citizen science. Multidisciplinary means that the application of citizen science covers several subject areas, that range from agriculture to hydrological sciences and water resources management to involve social sciences and humanities, as illustrated in the pie chart in Figure 1.

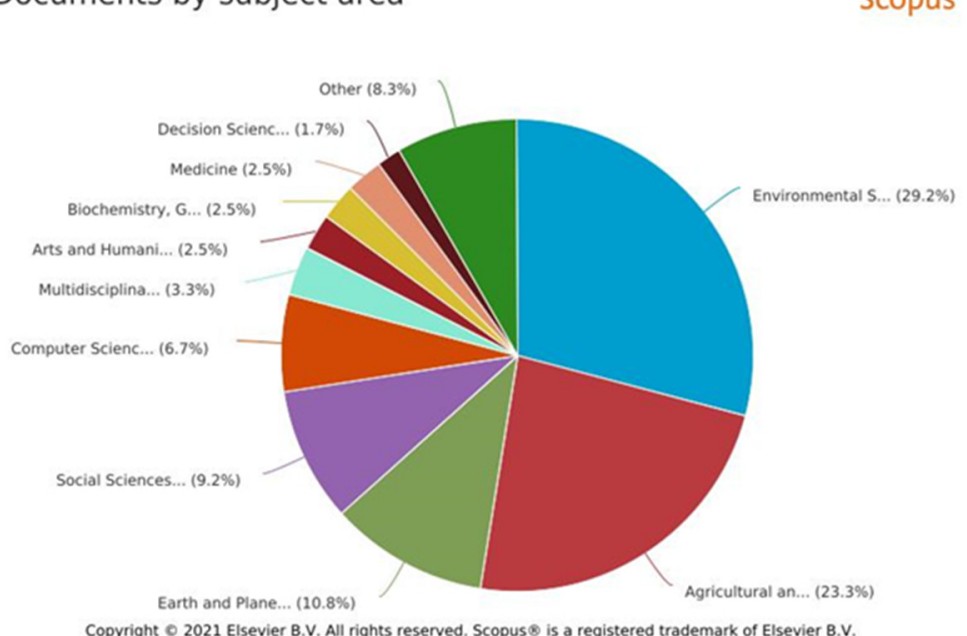

**Figure 1.** Documents by source based on 70 selected results (source: Scopus).

Although these multidisciplinary characteristic, environmental, agricultural and earth observation sciences are the most common topics that adopt citizen science approaches, a minority of publications refer to social sciences and computer sciences, as illustrated in the graph in Figure 1. Social sciences investigate on behavioral and motivational aspects at the base of community building (both physical and online). Social science also investigates into the cultural, jurisdictional and economic background in order to contextualize the citizen science environment of application for supporting qualitative research. While computer sciences are focused on finding technical solutions for citizen science projects, such as the use of social media, development of web platforms and user-friendly interfaces, supporting the quantitative analysis of data collection, with the integration of statistical methods. The residual percentages attest to a growing interest in citizen science towards disciplines in which the experimentation of participatory models is usually limited or null.

Transdisciplinary means that citizen science approaches integrate different scientific backgrounds and research design for science-driven solution-oriented applications to solve a specific problem that requires support and engagement of a wide range of potential non expert users and actors (e.g., stakeholders, industries, public agencies, general public, etc.) that are interested or impacted by the issue of interest [16]. For example, conservation studies or participative environmental and water resources management projects often require sociological skills to understand volunteers' behaviors within an organization. Similarly, citizen science approaches require skills in organization studies to understand the functioning of volunteer groups and the assignment of roles and tasks within an

organizational structure as well as IT skills for the development and implementation of suitable tools for participants' engagement, data collection and analysis activities. The selection procedure adopted brought to light conceptual, epistemological, technological and organizational issues related to the application of citizen science in empirical research contexts. The structure of findings in the next section was organized to report these aspects in detail, highlighting frequent opportunities and limitations in the application of citizen science.

## 3. Findings

### 3.1. An Overview on Definition of Citizen Science

Citizen science refers to the participation of the general public (non-experts) in the generation of new scientific knowledge [16,18–20] and in facing data scarcity, gathering meaningful information and addressing the lack of systematic coverage of extended or complex domains [14,15,21]. Non-expert participants' engagement is functional to challenge new environmental and scientific issues through a capillary monitoring usually insufficient if limited to experts [22], and to build new knowledge [23] based on a new collaborative and shared research design between expert scientists and citizens that implies a common determination of scope, scale and activities, such as research question definition, data collection and analysis methods, outcomes discussion and dissemination [6].

Citizen science constitutes an innovative research approach aimed at the integration of human and societal perspectives into scientific tasks [24]. Humans have a pivot role in the construction and definition of new research design, in the two-fold scope to: (1) investigate into bidirectional relationships between environmental and socio-economic systems [24]; (2) model new organizational processes supported by flexible interaction between human and computer [5,25].

Citizen science offers new possibilities to develop science through bidirectional trade-offs of information and knowledge between scientists, experts, stakeholders and volunteers in general or the public [26]. Scientists can connect volunteers' contribution and information to broader scientific literature. At the same time, contributors can guide scientists in data interpretation and discover common concerns to which science can give answers and solutions though a constant bottom-up observation of phenomena under investigation [26].

In this view, citizen science is a tool that coordinates citizens to action [27], combining scientific purposes with decision and policy-making processes under a community-based approach [24,28], also if it means including subjective element into scientific tasks [24].

Some authors propose a definition of citizen science similar to a creative crowdsourcing [29], while most of the studies agree with the assumption that citizen science occupies a superior level among the participatory approaches [30]. According to this latter vision, crowdsourcing is an important tool for citizen science but not a research approach at a whole.

A common and statutory definition of citizen science is still missing [30]. Most common definitions distinguish three typologies (or levels) of citizen science: (1) contributory, (2) collaborated, and (3) co-created [1,6,10,13,31]. The distinction of these three types depends on the level of engagement and tasks assigned to volunteers. (1) Contributory refers to projects designed by expert scientists, where volunteered contribution is limited to data collection [1,6,32,33]. Contributory citizen science can assume consultative function when authoritative entities—such as governmental bodies or institutions—support scientists in research question and research design definition [8]. (2) Collaborated refers to projects generally defined by scientists, in which volunteers can contribute to refine research design and activities such as data collection and analysis, and dissemination of outcomes [1,6,33]. In a socio-hierarchical perspective, stakeholders can replace or work alongside expert scientists in project definition [8]. (3) Co-created refers to the collaboration of scientists and volunteers from the co-design and co-conceptualization phase, where participants are actively involved in most or all stage of the scientific process [1,6,33]. Co-created citizen science can assume transformative functions when a local community leads a project [8].

In addition to this traditional schematization, another conceptualization of citizen science proposes: (1) contractual dimension, where a local community employs professional researchers to conduct specific investigations for common concerns [33]; (2) collegial dimension that implies independent research by volunteers submitted to professional researchers for a scientific review [33]. These latter typologies tend to strongly limit the role of expert scientists in the research design process [33]. Beyond this schematization (as illustrated in Figure 2), it is possible to assume that citizen science contributes to establish a constant dialogue between communities, stakeholders and experts and support the understanding and engagement of volunteers from local communities into scientific tasks [1,28].

| Typology | Research design | Volunteers' role |
|---|---|---|
| Contributory (consultative) | Projects designed by expert scientists *(Bonney et al., 2009)* or by authoritative bodies *(Ferster & Croops, 2013)* | Volunteers' contribution limited to data collection *(Bonney et al., 2009)* or to general consultation *(Ferster & Croops, 2013)* |
| Collaborated (collaborative) | Projects generally designed by scientists *(Bonney et al., 2009)* or by stakeholders *(Ferster & Croops, 2013)* | Volunteers' engagement extended to refine research design and the whole research activities from data collection to outcomes dissemination *(Bonney et al., 2009)* |
| Co-created (transformative) | Scientists and volunteers work together in all steps of research *(Bonney et al., 2009)* or local communities take the lead into projects *(Ferster & Croops, 2013)* | |
| Contractual | Local communities employ professional researchers for scientific investigation *(Green et al., 2020)*. Citizen Science connected to community-based approaches *(Commodore et al., 2017; Adler et al., 2020)* | |
| Collegial | Indipendent research by volunteers submitted to experts for a scientific review *(Green et al., 2020)*. Marginal extremist positions into scientific community *(Bonney et al., 2016)*. | |

**Figure 2.** Citizen science typologies summarizing table.

The conceptualization of citizen science implies a holistic perspective that concerns social inclusion, cultural differences and jurisdictional framework [31,33] functional to: (1) empower general public participation in research activities and raise public knowledge and awareness in a context in which citizen science is considered as a pillar of open science in political and socio-economic systems [34,35]; (2) contextualize and connect scientific research to social and institutional needs [36].

The debate on the definition of the concept of citizen science also focuses on the role played by digital and web-based technologies advances. The use of specific technological tools—such as collaborative database, social media, networking and collaborative technologies—undoubtedly constitutes an added and innovative value for the recruitment, involvement and training of volunteers in scientific research processes [20,21,25]. Citizen science represents an innovative approach for computer and data science in acquiring, integrating and modelling a massive quantity of data [5,37]. Nevertheless, the use of digital technologies does not necessarily constitute a fundamental requirement for citizen science, although it facilitates and enhances the application of its principles in the context of scientific research.

Differences between Citizen Science and Participatory Approaches

Participatory approaches indicate a set of ways of involving volunteers in scientific research from data collection to co-production and dissemination of results. In their board sense, crowdsourcing and citizen science are part of this set.

Citizen science differs from other participatory approaches in terms of the level of engagement of volunteers. In a citizen science approach, amateur volunteers are involved in the research design, data collection and interpretation processes together with experts [1,2,6,23,30,38]. According to the level of expertise or experience, citizen science can involve participants at various stages of the scientific process [12], while traditional participatory approaches tend to leverage volunteers, regardless of their own skills. Then, citizen science can modify organizational issues by diversifying tasks of participants with new forms of online or network participations [12,39]. Participatory efforts in a citizen science process are also addressed to build a community of volunteers that cooperate in coordination with experts to achieve specific research goals.

Instead, in traditional participatory approaches, the role of amateur volunteers is generally passive and limited to data collection under the supervision of experts. In some cases, data collection could be unintentional or unaware [15]. In traditional participatory approaches, volunteers usually act as sensors. The term "participatory sensing" indicates that the role of volunteer participants is limited to sensor activities in a context of research definition and designing under the supervision of experts [8]. In this context, volunteers are engaged just to collect data with the support of digital technologies or by standard surveys [25,40]. Experts not only supervise the design and goals of the research, but also the design technological framework and tools. An example of participatory sensing is strictly connected to Volunteered Geographic Information (VGI) [41], as a complementary, and in some cases unaware, information provided by volunteers or the general public by their own georeferenced personal devices. In the context of VGI, people act as a human sensor, providing information about geospatial context [41].

Citizen science and participatory approaches present similarities in searching for new ways of knowledge production [1,25]. In the achievement of new sources of knowledge, researchers and experts adopt strategies to increase public understanding and awareness and to make the whole scientific process more participative [25] thanks to the support of web-based technologies and the combination of scientific research and social inclusion purposes [25]. Research monitoring programs are usually approached from a top-down data-centric perspective [42]. Engagement of people with a different level of expertise and knowledge can contribute to enrich point of views in research activities and enlarge research perspectives [42] by paving the way to citizen science approaches that also consider socio-cultural perspectives [42].

Citizen science also differs from citizen observatories, as an approach for the observation of phenomena for earth and environmental monitoring [10,43]. The main difference between the two approaches lies in their own benefits and goals. If citizen science is an innovative approach to scientific research, citizen observatories is a supporting tool for governance and policy in a perspective aimed to create synergies and networks between citizens and stakeholders to solve local policy about environmental issues [43]. Despite their own different goals [43], citizen science and citizen observatories can have many similarities such as methods and tools that improve and stimulate participation and people engagement on the topic of common issues [10]. Citizen science can trust on the support of local communities, but it is not characterizing. Local communities, instead, are characterizing for citizen observatories. The latter can have general regulations but a different implication. Citizen science can have general regulations applicable in every context.

### 3.2. Volunteers' Engagement

3.2.1. Recruitment and Motivations

Recruitment and motivation are two key aspects in setting up citizen science projects. Volunteer recruitment simply consists of the act of finding and retaining non-expert par-

ticipants for a long-term period [8]. In a citizen science perspective, recruitment is aimed at building organized groups of volunteers to build active communities around specific research topics. Matching scientific purposes with social and local community needs can make the volunteer recruitment easier and stimulate collective participation and collaboration between volunteers and experts. Therefore, citizen recruitment opens up to new innovative ways to reduce the gap between non-experts and professional scientists and helps experts to produce knowledge for purposes that matter to the general people [24].

A community is a complex group that can involve different kinds of actors such as, (1) expert scientists in the role of coordinators, (2) decision makers (3) stakeholders and (4) the general public or citizens [5,42] that share a common vision of society and organization [42]. However, different typologies of participants within the community can imply a different level or stage of involvement within the participatory process according to expertise, experience, and personal interests [5]. The participatory process, in fact, is articulated in different steps: envisioning and goal settings, model formulation, data collection and cross-checking, model application and evaluation of outcomes [5].

Challenges in recruitment activities usually concern the lack of volunteer interest or of participant diversity [44] and the lack of familiarity with science, socio-political and jurisdictional framework limits [2,13]. Social interactions, frequent communication between participants and stakeholders, enjoying the outdoors and a frequent call to action are tools for long-term recruitment and keeping the motivation of participants high [6,20,27,28,45]. The frequent call to action and motivation can be stimulated by gamification of activities, feedback by email from experts, the use of social media and participation rates [2,5,33]. These tools not only stimulate communication between experts and non-experts, but also provide opportunities for social interactions within the participants' community, increasing their awareness and motivations [33]. Recruitment and motivational methods can be enhanced through partnering with existent organizations—such as civic groups, neighborhood organizations, target communities—that can support long-term activities [8].

Recruitment activities tend to include volunteer motivation in participatory processes, considering human behaviors, cultural and social attitudes and personal experience [3,10], even if these aspects can be at the origin of biases and the heuristic mechanism [5]. Understanding motivations and human behaviors is a critical area of investigation and represents an integrated part of research design in citizen science projects [16,25].

Motivations at the base of volunteer recruitment and engagement are: (1) personal interest [1,46]; (2) scientific knowledge for the better understanding of their environment or to gain political leverage [2]; (3) improving relationships between people, institutional actors and stakeholders [2,5] aimed at social learning and co-management of common resources and goods [47]; (4) promoting joint action and civic participation on environmental topics with socio-economic and cultural implication or that involves socio-organizational aspects [3,15,23–25,28].

Volunteers' motivations may vary according to their level of expertise or personal experience. Neophytes or amateurs with no expertise can be motivated by altruism, curiosity to a specific research topic, curiosity to challenging their capabilities, improvement in awareness, learning more about something familiar [10,25,48] or the opportunity to simply have fun [10]. People with a minimum level of expertise are, usually, motivated by a desire to share their knowledge, contribute to science (also only with data collection in support of scientific activity), join a community or contribute to social development [10,25,48]. Pride and prestige, especially, move to participation authorities or authoritative bodies [48].

Keeping high volunteer motivation implies a long-term strategy of engagement. To reach this goal, expert recruiters have to consider public gratification [6,48]. Volunteers must constantly feel that they have an active role within citizen science projects. A dashboard of data visualization, frequent call to action by email, sharing information with social media, gaming activities and ranking systems are fundamental tools for keeping high participation and motivation by giving constant feedback on volunteers' work and on their necessary role in achieving research objectives [2,5,6,10,25,33,48].

### 3.2.2. Community Building and Online Communities

Citizen science projects acquire more value when the scientific question has correspondence with local (environmental) issues and meet social and cultural odds [24,45,49]. In this perspective, volunteer engagement acquires a meaning around the sense of community and belonging. The interconnection between the community-based approach and citizen science is aimed at addressing research question and decision making to specific needs of communities [28]. Local community participation in citizen science activities helps to reduce social and environmental disparities and enhances the role of communities into research and decisional processes [28]. Community-based monitoring can be considered as a part of citizen science in which non-expert citizens, authorities, stakeholders and scientists can collaborate to respond on common concerns [50].

Social inclusion and dialogue between expert scientists and volunteers are two pillar characteristics of citizen science approaches. Experts usually care about the research design, recruitment, engagement, training of volunteers and finally the outcomes dissemination. The term "experts" usually indicates people with an academic or scientific background, able to conduct scientific research [2]. In this perspective, the category of experts does not include people with expertise or professional experience in a specific task. The latter composes the vast and heterogeneous group of volunteers, defined as amateurs. Initially qualified as a simple citizen with no academic qualification or expertise [8] or considered as a human sensor, thanks to the support of personal digital devices [41], advances in citizen science usually distinguish different groups of amateurs based on their expertise or engagement level: (1) neophyte, with no official background on the research topic and acts individually for altruism or curiosity; (2) interested amateur, with self-experience in the research topic or some training provided by experts; (3) expert amateur/professional, very familiar with the topic but external to academic and scientific context; (4) expert authorities, that is governmental or public agencies [30,48]. Discovering differences among amateur participants is fundamental for organizing tasks and assigning roles within citizen science projects.

The implementation of web-based technologies into citizen science approaches give new perspectives for engagement enlarging participation from physical to virtual spaces. Enhancing the role of online communities into citizen science approaches is a challenging topic. Social media and digital technologies, in general, are emerging tools to recruit, engage and motivate organized people in achieving a specific scope [25]. Social media may make communication between experts and volunteers easier and, among the latter, contribute to build communities by a constant data and information exchange [33]. Involving online communities into the citizen science approach implies a new form of participant organization around (1) remote work teams, (2) collaborative task management, (3) management of the dynamic division of labor, and (4) communication with large audience [35]. Therefore, engagement of online communities may present limits in terms of data validation and reliability of participants [51].

### 3.2.3. Crowdsourcing

Crowdsourcing is an activity to obtain data or information from a large number of people by the use of web-based technologies [5,29,40]. Crowdsourcing consists of the externalization of repetitive and simple tasks to the non-expert public, aimed at data collection from volunteers [9,19,40,49]. It could be considered as a valid tool for engaging people and obtaining data and information at a low cost and rapidly at real-time [29,40].

Crowdsourcing is an emergent tool in research activities aimed to cover the lack and scarcity of information on the strategic field as the environmental and geographic field, affecting the organization of volunteers' and attribution of tasks [14,40]. Constant availability of information is essential in designing and managing activities [14].

Including crowdsourcing as a method of citizen science, means that the latter has a multilevel definition regarding to the grade of participation and engagement [1,14]. Citizen

science refers to volunteers conducting science, while crowdsourcing refers to volunteers giving data and resources [19,23], but they are not involved in research activities [52].

Unlike citizen science, crowdsourcing activities do not necessary imply the active engagement of people, but can be conducted with a passive contribution from a sensors-installed network, the Internet of Things or internet technologies in general [14]. Crowdsourcing data are less accurate than official measurement tools but guarantee most spatiotemporal coverage that could be complementary [14]. If citizen science needs to recruit people to conduct its activities (from data collection to research design), crowdsourcing can only make use of unintentional contributions provided not only by citizens, but also by sensors and IoT [14,19].

Crowdsourcing activities mainly use two categories of tools: (1) a sensor installed by citizens with authoritative supervision; (2) mobile devices for either intentional or unintentional collection [14] as, for example, public social media content. In this latter case, the use of data is anonymous [11]. The use of technologies in citizen science, instead, is intentional and aware under an agreement between different users and stakeholders that take part in projects.

### 3.3. Tools

3.3.1. Digital Technologies

Digital technologies, usually, refer to mobile personal communication devices (MPCDs), such as a camera, microphone, GPS and data storage devices [8,10,33]. New digital technologies are tools for a low-cost, more geographical and time series coverage [38] and that complement traditional instruments of data collection [2].

Personal devices provided with a camera and GPS offer a new scale and perspectives of information [8]. The advances in digital technologies and consumer electronics offer new opportunities to gain data not only from experts, but from the general public as well, who can contribute to data collection and provision with their own personal devices that can guarantee a broader spatial and temporal coverage [8,53]. These advances contribute to create a distributed network of sensors spread around the geographic space and to constantly operate during time and at a different scale and perspectives [8,33].

Firstly, researchers use digital technologies in citizen science projects to facilitate and enlarge the participation of general users because of non-intrusive characteristic, commonly used in the daily life of citizens [9,12,54,55]. Secondly, digital technologies are catalysts for community-based participatory initiatives, enhancing communication between experts, stakeholders and citizens and organizations within local social networks and online communities [13,55]. On the organizational level, advances in digital technologies improve data collection methods set up on the direct handling and sharing of information [55]. The use of digital technologies also allows access to additional and auxiliary information, such as insights on real-time and location-based information as in the case of Volunteered Geographic Information (VGI) [6,41]. Furthermore, experts use digital technologies for the assignment of roles and tasks within the organization according to different levels of participation such as: data collectors, designers of problem definition and data collection methods, data framing interpretation and knowledge sharing [55].

3.3.2. Web-Based Technologies

The development of technological systems and web-based services offers new opportunities for a better engagement of volunteers by distributing, to them and to experts, infrastructural solutions for connecting devices and data storage [2,30,56]. Thanks to modern technologies, data can be collected on unprecedented scales even with limited funds [12].

The spread of internet technologies has changed the landscape of information availability and the way people use to interact and set attitude and behavior [11] and foster the development of participative activities around citizen science approaches [40]. The use of web-based technologies facilitates the interaction between people of different ages,

geographic origins and different motivation and goals [11]. These interactions, instead, pose some reflections about how technological tools can be used and pose some issues about relationship and human–computer interactions (HCI) [11,25] and other reflections on user experience (UX). In order to facilitate HCI and UX, the design is the first approach. Design must reflect cultural and environmental characteristics that introduce users into research activities and reduce the technological knowledge gap and, in any case, also availability and coverage [11]. Developing platforms for data sharing and visualization facilitates the retaining and motivation of volunteers [11]. HCI and new communication media are tools for facilitating stakeholder engagement and for enhancing the smart adaptability of organizational and decision processes by inserting human interaction into process modelling [5,25].

### 3.3.3. Social Media

The pervasiveness of social media is changing the way users interact and the ability to build social networks [18,57]. The successful design of internet technologies and social media platforms influences the way users interact and the capacity to create online groups of volunteers [12,18,57]. These aspects offer, first, new ways of data collection and implementing research models set up on a constantly shared and updated information exchange between experts, stakeholders and volunteers [5]. This pervasiveness makes social media platforms as interesting, potentially fundamental, tools within scientific research activities to reach and engage an increasing number of participants able to collect an unprecedented amount of real-time data, expanding the geographic coverage of information [58–60]. Similarly, social media platforms can be useful tools to simplify organizational steps of citizen science projects, such as recruitment and task assignment [12,57], outcomes dissemination and call to action [61]. Social media can also be used as mediation tools between experts and volunteers to understand participants' behavior within the social and environmental context of action [57,59]. In fact, social media can give direct and unfiltered information on aspects concerning both personal and collective attitudes and perceptions [59].

However, the use of social media data, in the context of citizen science, remains limited and experimental, due to the potential biases deriving from the cognitive perception of participants [5,59].

### 3.3.4. Gamification

If digital devices and smartphones are widely considered as consolidated tools for dealing with citizen science activities, gaming represents a new innovative frontier for long-term engagement of volunteers and keeping the motivation of participants high [61].

Gamification contributes to what is defined as "distributed thinking" [24] and is a tool for motivating volunteers in data collection and participation activities by designing interactive and dynamic graphic interfaces [11].

Serious games and gamification are two pervasive and innovative tools in the communication of science and in engagement of citizen scientists [44,62]. They reflect the emotional and motivational aspect of citizen scientists [62] and are promising tools for teaching concepts, distributing training and skills, communicating scientific topics and promoting participatory modelling [5,62]. The main potential of gaming is its leisure and fun characteristic that can stimulate non-expert participation [62].

Serious games encourage participants to evaluate their actions, raising awareness of the consequences by creating a simulated scenario and offering an engaging and challenging frame [62]. Simulated scenarios may be functional in understanding cognitive processes in the context of evaluation of alternatives and decision making [62].

### 3.3.5. Artificial Intelligence (AI)

Artificial Intelligence (AI) represents an innovative tool for advances in citizen science approaches. Reliability and accuracy of observations are important issues in data collection with personal devices. AI can contribute to data collection and validation by developing

algorithms for data comparison or sampling [33]. The use of artificial intelligence, drones and sensors can help to convert citizens' and volunteers' data into useful information by comparing VGI and user-generated contents (UGC) from camera and personal devices to official data [26]. AI supports the definition of data protocols and sampling and the improvement of citizen science efficiency.

Artificial Intelligence is frequently used in ecological and environmental citizen science monitoring to identify elements provided by volunteers in the form of digital data, such as images or videos [33]. The use of AI in citizen science monitoring aims to improve efficiency and classification accuracy by detecting volunteered observations and connect them to samples from traditional monitoring tools [33].

Developing AI solutions for reducing errors and biases of volunteers' contributions support validity of citizen science outcomes and data quality validation [33]. Overall, the integration of citizen science and AI technology can be used to help maximize the amount of data that can be collected and processed efficiently, while simultaneously engaging and informing people about professional research activities and their value for society and decision-making processes that affect socio-economic and environmental systems [33].

### 3.4. Data

#### 3.4.1. Data Types

Most challenges and opportunities of data collection in citizen science approaches are the need to have a larger spatial coverage and constant updating time series [2,38,40,44,53,63].

Citizen science implies the integrated use of data types from different sources to improve research models and support decision-making processes based on constant and widespread monitoring of phenomena: (1) official data, (2) volunteered data, (3) incidental data [64]. Official data refer to authoritative data produced by experts, such as researchers, professionals and technicians in the context of their professional or institutional activities, collected with specific tools under reward. Volunteered data are related to volunteers' activity coordinated by experts who provide guidelines for data collection in the context of citizen science in general for purely scientific purposes without rewards [3,64]. Incidental data refer to the whole crowd data available on the web. Incidental data usually refer to information—usually unstructured—furnished by amateurs for different purposes distributed in forum or social media groups and pages [64].

Platforms and visualization of results (1) enhance the reliability and efficiency of citizen science projects [2], (2) facilitate exchange and collaboration in modelling processes between different actors [5], (3) build joint scenarios and (4) formulate a shared hypothesis and research design [2,40]. In addition, the development of visualization platforms can facilitate the engagement and increase the long-term motivation of volunteers because of the immediacy and transparency of communication.

Developing data collection and visualization tools is fundamental for citizen science research approaches to obtain knowledge and useful information to support decision-making processes [37].

#### 3.4.2. The Role of Geographic Information: VGI and Participatory GIS

Among the different types and sources of data for citizen science, Volunteered Geographic Information (VGI) occupies a particular section, because of its specific characteristics about geographic information and the opportunity to find contribution in geospatial context. Advances in GIS and VGI with personal devices support a holistic participatory approach in research and decision-making activities [39]. VGI can facilitate the contextualization and the distribution of information linked to environmental and social issues, as a context of origin of contribution [41]. VGI and participatory GIS are typically useful in investigating human behavior in the geographic and environmental context [16,39,48,65].

Mapping activities with participatory GIS (PGIS) furnishes supplement data for a better understanding of geospatial context and factors [5]. PGIS facilitates the communication between the base of group decision systems, supports the creation and exchange of new

knowledge by analyzing impacts of geospatial factors in modelling processes, provides a transparent and user-friendly visualization of data and results with dynamic maps [5,39].

Volunteered efforts in geographic data collection and production have redefined the traditional roles of experts in mapping activities [41]. VGI in research is a hot topic related to the development and advance in web technologies for involving people and obtaining several types of data with geographic location which helps to contextualize the information available [48]. VGI contents are the pillar source in the earth citizen science framework [48], such as object identification, measurements, transcriptions and corrections of existing data, indications and complementary indications from volunteers [48]. VGI can be classified as: (1) automatic and implicit, when users provide data unintentionally by the use of software or application that use GPS o geolocation tools; (2) manual and implicit, related to social media, gaming and web behavior and trends in general; (3) manual and explicit, when volunteers are aware of the objective of their action; (4) automatic and explicit, when users are aware and furnish VGI by software or specific applications with GPS [48].

### 3.5. Outcomes and Opportunities

Evaluation of social learning and outcomes is a complex task. Socio-organizational and economic impacts of citizen science are not direct but may give significant feedback on the motivational aspect at the base of engagement. Citizen science has social and educational benefits in terms of raising knowledge, education and awareness around public interest topics [18,21,24,34,54,63]. Citizen science offers new methods and tools for the generation of scientific knowledge, motivation of public engagement with scientific research in common concerns [63] and provision of scientific education [18,21,54]. Citizen science represents a common space for socializing; it creates a sense of belonging that contributes to participants' performance and outcomes [46] by improving efficiency, effectiveness and the scope of research processes [25]. Citizen science creates a space for social inclusivity, dialogue between stakeholders and allows people to express themselves by giving a voice to non-experts [25], starting from personal and community interests related to a specific local issue, such as resource management or the optimization of decisional processes [12]. In this way, citizen science increases the research capacity and efficiencies of crowdsourcing, social knowledge, empowers communities and engages policy and decision making [9,29,34]. Social inclusion and empowerment of communities depends on the ability of research activities to be inclusive and open to local communities' perceptions and socio-cultural instances [9,25,27,45].

Social inclusivity—derived from citizen science activities—contributes to mitigate conflicts around environmental and resources management [12]. It also contributes to improve the economic situation of participants by giving them knowledge and tools for managing local issues [12]. Citizen science also contributes to the professionalization of volunteers into scientific context [46]. Finally, it promotes to reduce the gap between professional scientists and people outside the academic context [12].

Additional benefits of citizen science can be recognized into the detection of rare and unpredictable events [21] and the possibility of discovering something new accidentally [54]. These characteristics allow to build and constantly update data and procedure models that positively impact on organizational and logistical aspects of research and decision processes [21].

The benefits in citizen science are measurable into equitable decision making and the facilitation of dialogue in policy making, by making citizens aware, on the one hand, and taking into consideration the communities' proposals and problems on the other [2,15]. Citizen science represents an occasion of dialogue between citizens, local communities, stakeholders, institutions and scientists [28,45] in a vision in which communities address research questions, express concerns, while scientist offer an answer on the base of citizen contribution in covering data gaps, and authorities take decisions [49].

Citizen science projects can rely on the community-based approach, in order to guarantee social well-being, because it links community needs to the scientific framework and

gives voice to general people usually not considered into research activities [4,23,25,54]. In this perspective, citizen science gives the potential to generate social capital and take collective action [45], strictly connected to the context and place-based knowledge and social bonding [27]. Citizen science offers the opportunity and concrete context to enhance stakeholder participation by promoting communication and engagement activities coordinated by experts [27]. It means that citizen science can coordinate citizens' efforts in decision and policy-making processes [9,27]. Linking science with community needs can increase citizen participation and keep motivation high for a long term [4]. In this perspective, citizen science makes the research activities societally relevant [23,32] and develop large-scale and long-term monitoring under scientific supervision in order to support and plan decision-making processes [23].

Indirectly, citizen science helps social and economic research because it investigates behavior behind action taken [54]. Citizen science implies deep personal and social changes in behavior because it imposes a self-reflection and examination of belief systems at the base of socio-cultural organization systems and economic behavior [27]. Citizen science poses, also, deep reflections on the cultural context of application and general organizational issues in order to develop and promote a participation culture at a national or regional scale as a structural methodology for the monitoring and organization of environmental governance supported by bottom-up observations, by integrating data and expertise and the stakeholders' network [42]. Citizen science programs can enhance decision-making processes by stakeholders and governments [45] through the decentralization of organizational processes and new management towards public engagement and awareness [5,54]. An indirect benefit of citizen science is promoting social justice and equality and improving education around common concerns [26].

### 3.6. Limits and Challenges

### 3.6.1. Lack of Standardized Protocols

Citizen science presents limits and challenges mainly related to the lack of standardized and universal protocols, applicable to multiple case studies [24,30]. Research design and protocols often adapt to specific case studies or to the solution of single issues limited to restricted geographic or social contexts [63]. A lack of standardization in research design limits the possibilities of comparison between different case studies and the application of large-scale citizen science models [24,63]. These limits can generate problems in terms of methodological rigor, data collection, definition of activities and coordination between volunteers [30]. The absence of standardized protocols or a universal research agenda specific to citizen science can threaten the elaboration of a new solid relationship, practices and interaction [51]. Due to different models and points of view, methodological approaches hamper the development of a unique protocol or research agenda [51].

Human perception and socio-cultural background may affect data collection activities and the fulfillment of tasks assigned to the volunteers [5,13,24,34,53,59]. Levels of education or training and cognitive biases can threaten the validity and reliability of volunteers' observations [28,50]. Subjective perceptions and biases not only influence interpretation and dissemination of results [34], but also may have potential negative consequences in economic, human and environmental concerns [26,50].

Citizen science needs the provision of adequate infrastructures in order to support communication, (online) training [53], storage of data collected, to offer analysis and standardize program evaluation [6]. Appropriate technology helps citizen science projects. Internet and smartphones are fundamental tools to facilitate the participation, but they are not a warranty of data quality and training is needed for the correct use of these technologies in citizen science tasks [9,12,20,21,55]. The success of participatory approaches, as crowdsourcing and citizen science, does not rely only on technological advances, but also on the capacity to engage people and foster cooperation and coordination between participants and stakeholders around common community concerns [14].

Citizen science presents socio-economic challenges in terms of controversies, insufficient funding, and barriers to participation related to social marginalization or political and jurisdictional impediments [2,9,25,30,40]. Citizen science may usually involve just a specific segment of society, reducing and underestimating other groups that can offer several perspectives on a specific concern [24]. Some projects reflect same values and opinions and offers a distorted vision of the local community as a whole [24]. Equal social representation in citizen science activities affects volunteers' retention and contributes to keep motivations of participants high for long-term goals [24]. Marginalization usually takes origin from social, political and jurisdictional (such as digital divides) issues that limit the participation to specific groups and exclude others [2,28,55]. Funding issues constitute an additional marginalization factor that hampers not so much data collection, but especially training, monitoring and quality control [45].

Finally, other challenges derive from data sensitivity and volunteers' privacy typical of research activities in general [13]. The main issue regards the ownership of data and the way in which data can be used and who can use them [23]. These open questions have an impact into ethical concerns [58]. Digital and web-based technologies have radically changed relationships and interaction among volunteers in participatory approaches [11]. This poses a serious reflection about privacy concerns in research activities as a potential barrier to public participation [11].

### 3.6.2. Data Validation: Quality and Uncertainties

Data validation is a fundamental and crucial step for determining reliability in citizen science projects [5,24,30]. It consists in the definition of data quality related to accuracy and consistency of observations provided by volunteers. Data quality is strictly connected to skills and training of participants [24,53], and to the appropriateness of the scientific method adopted [45]. Errors in data collection, in fact, generally depend on (1) volunteers' biases, and (2) the lack of standardization in data collection or data structure. Data quality also depends on the support of metadata description [8,34].

Errors in participatory data collection activities can be aleatoric, when referring to statistical uncertainties or epistemic when uncertainties depend on the lack of knowledge [5]. Errors in data collection can be classified as (1) random, (2) systematic and (3) representative [9]. Random errors can be reduced by averaging methods [9]. Systematic errors are related to human biases and can be excluded when recognized [9]. This kind of error is common when data collection activity is based on social media platforms, where user activities can be deceptive or misleading (i.e., troll) [59]. Social media data can also constitute problems related to data silos and in merging and comparing observations [58]. Representativeness errors depend on technical observation methods and tools [9]. Uncertainties can depend on a lack of participants' objectivity derived from subjective issues or by the adoption of inaccurate measurements [44]. Another source of errors can be derived from the digital divide that hinders data transmission and causes irregular intervals in time-series and geospatial coverage of citizens' contributions [2,14,44]. The availability of strong and widespread digital infrastructures is fundamental for volunteered observations' validation and reliability. The availability of a digital infrastructure is predominantly spread in urban and suburban areas, but less in marginal and rural areas. This issue implies that volunteered observations can be focused on a specific context and make the application of citizen science difficult at a large scale. Weakness in internet connection or GPS signal implies the lack of balanced and well-spread information or approximate and fragmented data where the internet connection or GPS signal is weak [8,44].

Challenges in data validation regards spatial and temporal resolution, cost, accessibility, availability, uncertainty and dimensionality [14]. Definitions of specific paradigms is useful for providing a priori principles for citizen science activities and data validation. Some factors, such as the spatio-temporal domain, socio-cultural organization or jurisdictional framework, can influence the adoption of paradigms and practices [36].

Definitions of ex ante or ex post sampling methods and standardization protocols constitute general solutions to deal with data validation [10,66]. Ex ante strategies imply the adoption of standard metadata, automatization of data collection tools, controlled vocabulary and adequate training of volunteers. Adoption of standardization models and protocols for data collection are functional to define a specific quantitative research question and address volunteers in data collection [6,20,45,63,67]. The establishment of a paradigm as the term of comparison for data quality and validation through sampling schemes aims to reduce errors and biases [8,28]. The adoption of a data model, such as predefined classes, could help to obtain complementary and useful information to integrate into models at the base of research activities and decision-making processes [40]. Standard ex ante models permit replication, community checking, training, profiling, reputation and finally discard outliers [40].

Ex post strategies are pivot around ranking volunteers' contribution, data mining, enrichment and comparison with external knowledge and professionals [12,48]. In this latter case, the adoption of comparison models or quality indicators as ISO standards facilitates data validation procedures [48]. Online data forms speed up the process of sampling and validation [1,66] by enhancing the cooperation between volunteers and experts aimed at solving data harmonization and homogenization issues using web 2.0 technology [21,64] and by the adoption of validation methods elaborated by experts [53]. Another validation strategy consists in gathering a large amount of data in order to obtain several terms of comparison for data sampling and research of standard protocols [37,40]. Gathering large amount of data includes not only the quantity, but also a wide geographic scale of collection [18]. Data enrichment passes also through the comparison of coupled data and data interlinks [46]. Gathering a large amount of data requests the involvement of distributed networks of volunteered data collectors [46]. Their contribution is important not only for gathering a large amount of data, but also to gain complementary information for the contextualization of information into social, cultural and economic frameworks [46].

Another possible solution for data validation is randomization [63]. Randomization consists in generating data representative for a specific field, period or location in order to obtain a sample of information [63]. Randomization allows research to address a general question and design a sample scheme [63] in order to verify quality and outcomes [6]. Standardization protocols, randomization and gathering vast amount of data help to build validation models through comparison and integration with authoritative data [64]. In this way, data interpolation and simulations are two common methods for testing data reliability and quality [2,9].

Data quality validation implies the adoption of a transdisciplinary approach, including social scientists—in order to investigate behavioral and background context—and statistical and computer scientists for protocol definition and data sampling [28,37,68]. New statistical model and visualization tools could represent a solution for enhancing data quality [24]. Integrating computer science into citizen science approaches helps to make efficient data entry and elaboration activities [24,28]. Especially, introducing statistical weighting is functional to rank volunteers and weigh their contribution [23]. Statistical models help also to find random errors or statistical biases [23]. Procedures for enhancing data quality and validation within a citizen science project include the supervision of experts and cross-checking activities among citizen scientists [23,68].

## 4. Discussion

The meta-review points out a variety of approaches and methods adopted in citizen science. The transdisciplinary application of citizen science provides a wide range of methods as useful insights for overcoming the single disciplinary fields towards an integration of knowledge and research tools within a general theoretical framework. The transdisciplinary framework constitutes an advantage, also than a necessity, for investigating many implications of citizen science. This implication regards computer scientists for technological structure; social and behavioral scientists for investigating relationship between

stakeholders and participant and their cultural and environmental background; experts on organization studies for investigating the organization and recruitment of volunteers and to analyze changes in co-production and co-design processes; then, experts on the specific field of applications (biologists, hydrologists, ecologists, geographers and so on. . . ).

These aspects emerge the importance of organizational studies and the potential role of online communities in carrying on a citizen science project by taking advantage of the wider spread and availability of digital and personal mobile sensors. The pandemic situation of COVID-19 also imposes a serious reflection on finding a new and innovative model to engage, motivate and organize people in giving contribution about common concerns—such as climate change, environmental challenges, the Water–Environmental–Food–Energy (WEFE) Nexus, resources management, monitoring and sustaining urban planning and services or reaching, at local scale, the Sustainable Development Goals (SDGs) adopted by the United Nations by promoting a collective call to action. Concerns that threaten our societies and have a strong impact on socio-economic systems, productions (industrial, agricultural) and cultural lives and wellbeing.

This review also highlights the evolutionary characteristic of the concept of citizen science. Starting from an effort of democratization on science through data collection, most recent perspectives enlarge the engagement to all the steps of research activities, from data collection to outcomes dissemination. The heterogeneous nature of volunteers and different motivations and behaviors at the base of their recruitment, make for more complexing citizen science activities, diversifying tasks and goals.

In a definition of a conceptual framework of citizen science, three pivotal points are fundamental: (1) data collection, (2) behaviors investigation and (3) building new models. The adoption of data collection methods aims to broaden the time and space coverage of available information, integration of data models and finally obtaining feedbacks from citizens. Behaviors investigations shed light into volunteers' background at the base of their choice and attitudes. Behaviors investigations may also shed light onto impacts of the decision and organizational model, both on the individual and societal level. Finally, citizen science approaches allow the investigation on relationship and integration between socio-economic and environmental systems for a new management of resources and optimization in terms of sustainability. Citizen science also allows finding shared solutions with the adoption of scientific methods for common and societal concerns. It gives constant feedback on adopted solutions and research design through an innovative organizational framework that includes scientists, stakeholders and citizens.

This contribution has shed light on key aspects of citizen science, as a participatory and inclusive bottom-up model on the co-design and co-production of knowledge. The heterogeneity of citizen science inputs and practical applications allow to identify a series of general requirements useful for the setting of participatory and inclusive models, with a view to continuous and constructive collaboration between experts and society.

Firstly, collaboration between experts and volunteers is based on the definition of a shared research question. A research question should intercept social needs and be oriented towards the solution of empirical problems. Researchers usually play a predominant role in this respect, for obvious reasons, related to the level of expertise.

Secondly, this contribution highlighted the varied typology of participants in citizen science processes. Each volunteer can be motivated by particular motivations and interests, and be the bearer of different and conflicting social and cultural demands. Researchers and volunteers should, therefore, work together to find a synthesis of views developing a common strategy towards targeted and specific objectives. They should, thus, identify defined target groups and set organizational models based on the characteristics, motivations and competences of the different groups of participants. Inclusiveness and bottom-up participation are two pillars in citizen science.

It does not mean that roles within an organization are necessarily equal and interchangeable. The role and position of each participant must reflect personal knowledge, skills and abilities. An organization active in citizen science includes a structure of roles

divided between experts and volunteer participants. The latter, however, can benefit within the organization in terms of professionalization, acquisition of skills and awareness, through the possibility of continuous training, dialogue and open discussion with experts and stakeholders.

Thirdly, this contribution highlighted the advantages of digital technologies for active and inclusive bottom-up participation, which can be measured, above all, in: (1) a greater spatial and temporal coverage of observations; (2) the ease and speed in communication, recruitment and internal organization between experts and volunteers. Based on these points, experts should rely on user-friendly digital technologies, widely used in everyday practices of citizens, such as smartphones and tablets. However, experts should provide a training phase for the correct use of digital technologies, appropriate to research objectives. The creation of graphical interfaces is also an important aspect to improve the user experience, facilitating interactions between participants through digital platforms for sharing and displaying data. The development of digital platforms for displaying data and results is not a secondary aspect, but an effective tool to increase the participation and sense of belonging of volunteers to the citizen scientists' community. In this way, volunteers become aware of their actions and the importance of their role in a collective problem-solving effort in scientific research.

Prescriptions, here illustrated, refer to a general context of application of citizen science. Based on these prescriptions, it is possible to design scenarios based on the schematization of citizen science reported in Section 3.1. Contributory, collaborated and co-created typologies can represent ideal-types for the definition of standard protocols, applicable and replicable to a different empirical cases study. From a contributory perspective, researchers and experts conduct all phases of the research. This implies that the adoption of citizen science as a research method is functional to solving a scientific gap. The role of volunteers is limited to the guided collection of data for the purpose of consulting and refining the research phases. Roles and tasks are predominantly homogeneous, with no distinction of expertise or background among volunteers. The social function of research questions may be secondary in such a context. The development of digital technologies is functional to the guided data collection activity to support the subsequent processing by experts who independently take care of the publishing and dissemination phases. The engagement of online communities is limited to the sharing of data mainly aimed at the collection of a large number of observations in the form of digital contribution.

Collaborated and co-created ideal-types indicate a more active involvement of volunteers in the implementation of citizen science processes. In the first case, the volunteers redefine the research activities set by the experts; in the second they act as peer with the experts. In this context, the definition of research questions assumes social importance, intercepting scientific gaps with needs expressed by communities, both local and virtual. Volunteers not only collect data, but define together with experts the methods and purposes of collection, in order to identify scientifically based responses to social and environmental problems. The definition of tasks and roles within the organization becomes more complex, as it must take into account the inhomogeneities among volunteers in terms of expertise, skills, motivations, interests and personal experience. At the organizational level, the structures of the communities involved become more articulated. Researchers maintain a shared coordination role with advanced volunteers who lead and educate a base of generic volunteers predominantly engaged in data collection activities. In this context, volunteers can increase their conceptual knowledge and technical skills, changing their status within the community. Such ideal-types imply changes not only in organizational, but also in behavioral terms, because volunteers and experts can contribute to the achievement of research goals through cooperation. It implies changing in individual behaviors because participants need to synchronize their action in order to create a group identity.

In a co-created perspective, participants are frequently involved in collective actions. It requires that participants act as a single entity in achieving a specific goal. Decisions

are binding and individual behaviors must reflect a general attitude. Participants not only share information and awareness, but also responsibility of their action.

Contractual and collegial citizen science typologies offer, instead, experimental elements with more uncertain outcomes. Firstly, the role of the experts appears to be limited only to the final review phase. The coordination of research activities is carried out almost exclusively by volunteers. In this context, organizational and technological aspects are difficult to place in a specific and functional structure because of the experimental nature of these approaches.

## 5. Conclusions

The aim of this contribution was reviewing main concepts, methods, tools and issues on the application of citizen science to lay the foundation for a theoretical framework across environmental disciplines, social and organization studies. In this sense, theoretical analysis through a systematic meta-review of literature has allowed to examine the multifaceted and transdisciplinary aspects related to citizen science. In addition to reviewing concepts, methods and tools, the analysis shed light on the possibilities that citizen science offers knowledge on co-production and problem-solving processes based on participatory approaches. However, theoretical analysis, also, presented the limits of this approach both from an organizational and behavioral perspective and from a technological perspective. At the base of these limits, there are mainly cognitive biases of participants, technological deficiencies, inadequate training, a lack of expertise and experience. This contribution, therefore, identifies pillars on which to design a theoretical framework of citizen science, valid for the management of participatory approaches and changes within organizations. These pillars can be summarized as: (1) definition of a research question that intercepts collective needs to provide scientific valid solutions to social, environmental and economic concerns; (2) identification of target groups active in a specific context; (3) analysis of behaviors of the volunteers and the motivations underlying their participation to assign roles and tasks appropriate to the level of expertise and training of individuals; (4) definition of standard protocols agreed between experts and volunteers to manage the research steps; (5) continuous interaction between experts and volunteers for the sharing of results and the implementation of any corrective measures in scientific co-production processes. Advances in digital technologies and in social media platforms are redefining methods and spaces for interaction and participation, expanding recruitment and engagement of volunteers in the context of online communities. These aspects offer ideas for further research on the topic of citizen science in relation to the changes and challenges posed by digital transformation. The conclusion reported here must be considered as a reflection on the sidelines of a theoretical review on the topic of citizen science. In this study, no empirical cases study was examined or conducted for the validation of statements. The lack of empirical validation represents a limitation of this study. In this sense, this contribution is limited to presenting a transdisciplinary theoretical conceptualization of citizen science as a starting point for the integration of methods and tools of environmental sciences in an analytical perspective linked to social sciences and organization studies.

**Author Contributions:** Conceptualization, A.S., S.G. and F.N.; methodology, A.M.B.; data curation, A.S.; writing—original draft preparation, A.S.; writing—review and editing, S.G., A.M.B. and F.N.; supervision, S.G., A.M.B. and F.N.; funding acquisition, F.N. All authors have read and agreed to the published version of the manuscript.

**Funding:** A.S. and F.N. acknowledge the support received by the WARREDOC center of University for Foreigners of Perugia through the grant agreement awarded to WARREDOC from the Istituto Superiore per la Protezione e la Ricerca Ambientale (ISPRA), grant number n. 123/2020. F.N. acknowledges the support received by the Southeast Environmental Research Center in the Institute of Environment at Florida International University.

**Acknowledgments:** Reviewers and editors are acknowledged for their valuable comments that helped authors in significantly improving this manuscript.

**Conflicts of Interest:** The authors declare no conflict of interest.

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
