# Peer review of "Towards a Transdisciplinary Theoretical Framework of Citizen Science: Results from a Meta-Review Analysis"

_sustainability, doi:10.3390/su13147904_

Round 1
Reviewer 1 Report
Dear Authors,
Thank you for your manuscript, whose aim is to contribute to the theoretical debate on a very relevant and timely topic.
My main suggestion about your article is to extend the considerations about the research methodology and design and, consequently, to re-arrange the structure of the subsequent paragraph to give more “order” to the paper. Indeed, in its current form, the reasons why you dedicate Section 3, 4, 5, 6 and 7 to citizen science, volunteers’ engagement, tools and data, outcomes and opportunities, and limits and challenges, respectively, are not so clear.
I would clarify this issue in Section 2, then I would include one “big” paragraph entitled “Findings” where, in an introductory section, to briefly summarize the structure of this part, and then to include in different sub-paragraph the main results of your theoretical analysis (what you now include in Section 3-7, in my opinion currently too long).
Finally, I would separate Discussion (where you should interpret and describe the significance of your findings) from Conclusions (where you should answer to the research question stated in the introduction).
Best wishes.
Reviewer 2 Report
I want to thank the author for submitting the paper “Towards a Transdisciplinary Theoretical Framework of Citizen Science: Results From a Meta-Review Analys". Even though I find the research idea and the topic in general quite interesting and relevant, I have some minor concerns that should be addressed. I will outline these concerns along the paper.
- There are serious flaws in terms of conceptual design of the paper - you cant’s rush into empirical part of your research without proper theory building. It makes your paper hard to follow. Generally, more theoretical insights and empirical evidence is needed for justification of your framework. Existing gap(s) are not clearly specified in the introduction.
- I would advise to focus on a description of systematic literature review basing on: Kraus, S., Breier, M., & Dasí-Rodríguez, S. (2020). The art of crafting a systematic literature review in entrepreneurship research. International Entrepreneurship and Management Journal, 16, 1023–1042. The work you have done is an interesting one, but you need to polish 2. Research Design more - eg. include inclusion/exclusion criteria of your SLR.
- Additionally, research limitations and future research avenues are not addressed at all.
Conclusions part require some work as well.
Reviewer 3 Report
The article is synthetic but cursory. The citizen science is very important discipline in the present-day world, but this text is like an essay, an excerpt from an introduction or coursebook. It contains too many general postulates and not enough concrete case studies. In my opinion is not scientific.
Round 2
Reviewer 1 Report
Thank you (and congratulations) to the authors for well revising the manuscript, which has consistently improved with respect to the previous version. In my opinion, it can now be accepted for publication in its current form. Best regards
Reviewer 3 Report
I would like to thank the Authors for the revised article. The structure is better. The overview on definition of citizen science is useful for readers. The authors pointed out involving of volunteers and lack of standardized protocols. I accept the text in this version.